# BILCO: An Efficient Algorithm for Joint Alignment of Time Series

**Xuelong Mi**[1], **Mengfan Wang**[1], **Alex Bo-Yuan Chen**[2], **Jing-Xuan Lim**[2],
**Yizhi Wang**[1], **Misha Ahrens**[2], **Guoqiang Yu**[1]
[1]Dept. of Electrical and Computer Engineering, Virginia Tech
[2]Howard Hughes Medical Institute, Janelia Research Campus
[1]{mixl18,mengfanw,yzwang,yug}@vt.edu
[2]{chena,limj2,ahrensm}@janelia.hhmi.org

## Abstract

Multiple time series data occur in many real applications and the alignment among
them is usually a fundamental step of data analysis. Frequently, these multiple time
series are inter-dependent, which provides extra information for the alignment task
and this information cannot be well utilized in the conventional pairwise alignment
methods. Recently, the joint alignment was modeled as a max-flow problem, in
which both the profile similarity between the aligned time series and the distance
between adjacent warping functions are jointly optimized. However, despite
the new model having elegant mathematical formulation and superior alignment
accuracy, the long computation time and large memory usage, due to the use of the
existing general-purpose max-flow algorithms, limit significantly its well-deserved
wide use. In this report, we present BIdirectional pushing with Linear Component
Operations (BILCO), a novel algorithm that solves the joint alignment max-flow
problems efficiently and exactly. We develop the strategy of linear component
operations that integrates dynamic programming technique and the push-relabel
approach. This strategy is motivated by the fact that the joint alignment max-
flow problem is a generalization of dynamic time warping (DTW) and numerous
individual DTW problems are embedded. Further, a bidirectional-pushing strategy
is proposed to introduce prior knowledge and reduce unnecessary computation,
by leveraging another fact that good initialization can be easily computed for the
joint alignment max-flow problem. We demonstrate the efficiency of BILCO using
both synthetic and real experiments. Tested on thousands of datasets under various
simulated scenarios and in three distinct application categories, BILCO consistently
achieves at least 10 and averagely 20-folds increase in speed, and uses at most 1/8
and averagely 1/10 memory compared with the best existing max-flow method.
Our source code can be found at https://github.com/yu-lab-vt/BILCO.

## 1  Introduction

Time series data appear naturally in a wide range of fields, such as motion capture, speech recognition,
and bioinformatics. As sequences are usually not directly comparable due to possible delay and
distortion, alignment among them is a prerequisite step before further analysis. The most popular
alignment technique, dynamic time warping (DTW) [3], finds an optimal match between two
sequences in linear time using dynamic programming (DP). However, DTW and its variants [2, 15]
only compare a single pair of time series, while real data often contain structural information. For
example, in time-lapse microscopy data, the pixel intensities are recorded over time in a 2D grid, and
adjacent pixels tend to have similar temporal patterns. Thus, joint alignment of more than a single
pair is expected to achieve better performance by taking advantage of dependency from the structure.

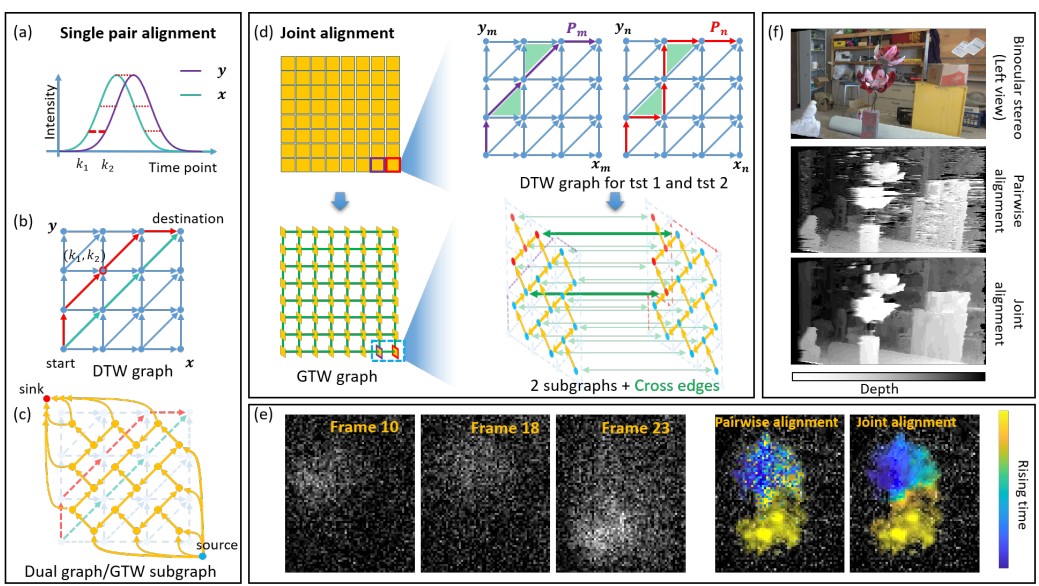

Figure 1: (a-c) The alignment between sequences $x$ and $y$. DTW graph of alignment is shown in (b) and its dual graph in (c). In (b), if warping path crosses $(k_1, k_2)$, $x[k_1]$ and $y[k_2]$ are matched. Each warping path in (b) is dual to a corresponding cut in (c). (d) GTW graph construction for time series arranged on a 2D grid. Details are shown for two adjacent pairs $(m, n)$. Given warping functions $P_m$ and $P_n$ (top panel), their corresponding cuts segmented the dual graph into the sink side (red) and source side (blue) (bottom panel). $dist(P_m, P_n)$ (the green shadow in the top panel) is proportional to the number of cross edges linking source side and sink side. (e) application of pairwise alignment and joint alignment to calculating signal propagation. (f) application to extracting depth information (data from [16]).

For a long time, it was not clear how to incorporate the structural information in a principled way. Various heuristic tricks were developed [13, 19]. In a recent theoretical breakthrough, the joint alignment problem was elegantly modeled as a max-flow problem of graphical time warping (GTW) graph by optimizing both the time series pairwise similarity and the distance between warping functions [22]. GTW was shown to have superior alignment accuracy in many applications, such as brain activity analysis [21] and liquid chromatography-mass spectrometry (LC-MS) proteomics [23], where warping functions represent the delay of the real signal at each time point relative to the reference time series and adjacent time series should propagate similarly. The binocular stereo vision formulation in [11] is also a special case of GTW, where warping functions denote the depth information contained in two views and it is natural to assume the depths are close in adjacent positions. By adding penalties on warping function dissimilarity, GTW utilizes the structural information and could obtain better alignment performance, as shown in Fig.1(e)(f). However, the long computation time and large memory usage severely limit its potential broad applications, because GTW was solved by existing general-purpose max-flow algorithms and the graph is often huge. Indeed, for a typical dataset with 5000 pairs of time series, if each time series contains 200 time points, all popular methods, such as incremental breadth-first search (IBFS) [8], Hochbaum's pseudoflow (HPF)[10], Boykov-Kolmogorov (BK) max-flow [4, 12], and highest-label push-relabel (HIPR) [9, 5], cost hours and more than 100 gigabytes memory.

In this paper, we identify two important properties of the joint alignment max-flow problem and show that they can be leveraged to design a novel algorithm with improvements in both speed and memory efficiency. Specifically, first, joint alignment is a generalization of pairwise alignment and numerous individual DTW problems are embedded, as shown in Fig.1(a)-(d). If the dependency is neglected, it is reduced to multiple independent DTW problems, which can be solved in linear time through DP. Although the dependency makes the problem more complex, the property of the DTW problem can still be utilized. Second, a coarse approximate solution to joint alignment can be readily estimated in many applications. Such prior knowledge can be incorporated to accelerate the max-flow algorithms.

Taking advantage of these two properties, we develop BIdirectional pushing with Linear Component Operations (BILCO), a novel algorithm that solves the joint alignment max-flow problem exactly and

more efficiently. The algorithm consists of two major strategies: (a) By utilizing the first property, we propose Excess pushing with Linear Component Operations (ELCO) that integrates DP and the push-relabel approach [9]. Each component is defined as a small subgraph including connected nodes and their related edges. The components are automatically and adaptively determined. We show that all operations on each component can be achieved in linear time by combining DP and subgraph properties. With linear component operations, ELCO can achieve higher efficiency than the generic push-relabel approach. (b) By leveraging the second property, we design a bidirectional-pushing strategy to utilize prior knowledge as initialization. The strategy simplifies the original problem into two smaller sub-problems with opposite pushing directions. In each sub-problem, ELCO is applied, but the separation of two pushing directions dramatically reduces unnecessary computation.

BILCO has the same theoretical time complexity as the best popular methods such as HIPR, but it provides a significant empirical efficiency boost without sacrificing the accuracy in the task of joint alignment. Its effectiveness is evidenced through both synthetic and real experiments. Compared with IBFS, HPF, BK, and HIPR on thousands of datasets under various simulated scenarios and three real applications in distinct categories, BILCO improves the speed by 10-50 folds and costs averagely $1/10$ memory relative to the best peer method.

## 2   Problem formulation

The joint alignment problem can be formulated as [22]:

$$\min_{P_n, n=1,2,\ldots,N} \left( \sum_{n=1}^{N} cost(P_n) + \kappa \sum_{(m,n) \in Neib} dist(P_m, P_n) \right) \tag{1}$$

where $P_n$ denotes the warping path for the $n_{th}$ time series pair, $N$ is the total number of time series pairs that are jointly aligned, $cost(P_n)$ is the alignment cost of the $n_{th}$ time series pair, $dist(P_m, P_n)$ is the warping path distance defined by the area of the region bounded by $P_m$ and $P_n$. $Neib$ is the set of pair indices $(m, n)$ representing the adjacent time series, and $\kappa$ is the hyperparameter to balance the alignment cost term and the distance term. For example, for 2D grid time-series data, $Neib$ may include the pair of neighboring pixels, and $\kappa$ represents a prior similarity between the pixels.

As shown in Fig.1, Equation (1) can be converted to a flow network and solved by finding the min-cut of it. The constructed graph, GTW graph, consists of $N$ GTW subgraphs $\{G^n = (V^n, E^n)|n = 1, 2, \ldots, N\}$ and cross edges $E_{cross}$ with capacity $\frac{\kappa}{2}$ (Fig.1(d)). The edges within GTW subgraph are called $E_{within}$. For convenience, in the following content, we refer to "GTW subgraph" as "subgraph". Each subgraph $G^n$ is dual to a DTW graph, which represents the warping between a pair of time series. The cut within a subgraph is dual to a warping path of the time series pair [1] (Fig.1(b)-(c)), and thus the min-cut of one DTW graph can be solved in linear time through DP by finding the shortest path. The cross edges constrain the difference between cuts in neighboring subgraphs, corresponding to the distance term in Equation (1) (Fig.1(d)). To ensure the monotonicity and continuity of warping paths, the capacities of reverse edges in each subgraph are set infinite [22].

It is worth noting that GTW framework is flexible, broadly applicable, and can be used as a building block in solving many problems while utilizing structural information. The assumed neighbor structure and similarity strength can be application-specific or user-designed. For example, for different pairs of warping functions, GTW can set different hyperparameters $\kappa$. For simplicity, in this paper we use the same hyperparameter $\kappa$ for all warping function pairs in the following.

## 3   Method

BILCO contains two major parts, ELCO and bidirectional pushing, as discussed in Sections 3.1 and 3.2, respectively. Based on the initialization of warping functions, the bidirectional-pushing strategy converts a max-flow problem into two sub-problems and we solve each of them with ELCO. The integration leads to BILCO (Section 3.3), followed by the analysis of time complexity and memory usage in Section 3.4 and 3.5. All lemmas and theorems are proved in the supplementary.

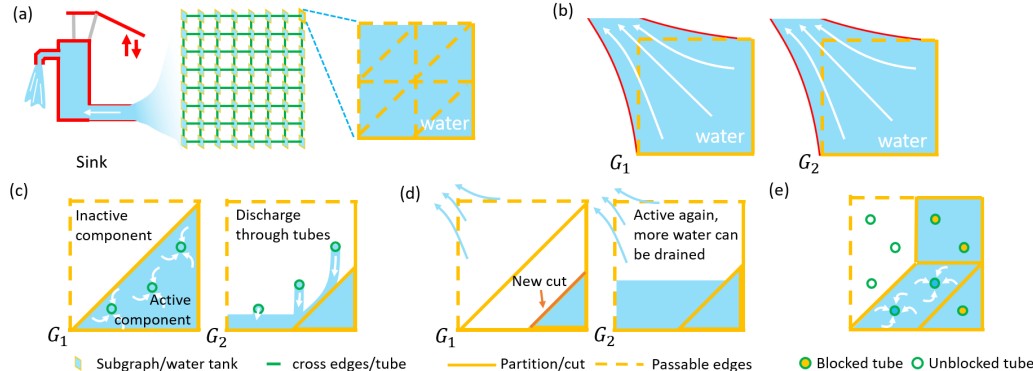

Figure 2: An analogy to joint alignment max-flow problem. Water, tube, tank, and partition represent excess, cross edge, subgraph, and cut, respectively. (a) In the initial stage, the source has pushed all water it can to water tanks. (b) "Drain" operation: water in the tanks is drained to sink. (c) "Discharge" operation: water flows from an active component to a neighbor component in another water tank, where water needs to concentrate at the connecting tube first. (d) After (c), the previously inactive component in $G_2$ becomes active and water can be drained again. Due to the concentration step in (c), there is one new partition in $G_1$ that further segments it into three components. (e) One intermediate stage of a tank, where multiple components are segmented in a subgraph and some tubes are blocked.

## 3.1 Excess pushing with linear component operations (ELCO)

Since GTW graph contains many subgraphs dual to DTW graphs and each DTW graph can be linearly solved by DP, we hope the DTW graph and DP algorithm can be exploited in our approach. While the cross edges between subgraphs prohibit the direct use of DP, we found that the component as defined in Subsection 3.1.1 is the maximum subgraph that can allow the use of DP. By combining the newly designed graph conversion strategy (Fig.3) and DP, we establish that the operations on components, "Drain" and "Discharge" (defined in Subsections 3.1.1), can indeed be linearly implemented as in Subsections 3.1.2 and 3.1.3. To guarantee global optimality, we design a new labeling function as in Subsection 3.1.4 and use that to guide our component operations.

### 3.1.1 The basic unit of operation is component, not node or subgraph

To exploit the property of each subgraph, ELCO borrows the idea from the generic push-relabel algorithm, whose operations are localized. ELCO allows the existence of excess, which is the surplus between the entering flow and outgoing flow of one node. The nodes that carry excess are regarded as "active". Based on the location of flow exchanges, we define two operations, "Drain" and "Discharge". "Drain" pushes excess to the sink through $E_{within}$, which is a within-subgraph operation. While "Discharge" is a cross-subgraph operation that pushes excess to neighbor subgraphs through $E_{cross}$. By alternatively executing these two operations, more and more excess can be sent to sink (Fig.2). The spread of excess leads to many edges being saturated, resulting in multiple cuts in the same subgraph. These cuts segment the subgraph into different components (Fig.2(e)). Formally, we define one component as a subset of GTW subgraph bounded by two adjacent cuts. ELCO uses component, instead of node or subgraph, as the basic operation unit, because it is the largest possible unit to use DP to push flows.

**Lemma 1:** If two nodes are in the same component, then there is at least one path linking them.

By Lemma 1, if one node in a component is active, the excess can spread anywhere in the same component. Thus, the whole component can be seen as one unit. On the contrary, cuts block the flow across components in the same subgraph, it is inapplicable to take a whole subgraph as one unit.

Using the component as the basic operation unit, ELCO can be described as follows: Initially, each subgraph is one active component. "Drain" operation sends maximum excess directly to the sink through $E_{within}$. A new cut will be generated, which blocks some excess in the segmented components near the source. Then it is the turn of "Discharge" to seek opportunities to move excess across subgraph toward components that can execute "Drain". By executing these two component operations alternatively, the global max-flow can be achieved. An illustrative example is shown

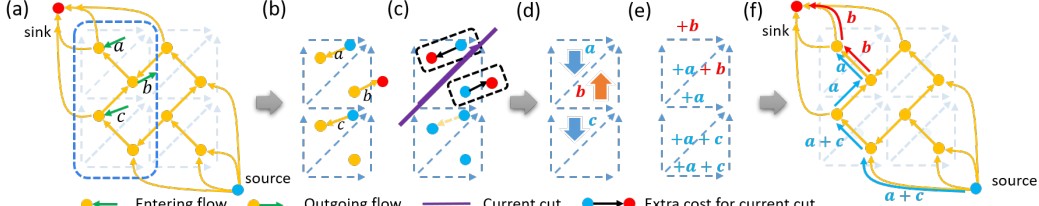

Figure 3: An example of graph conversion to incorporate flows on $E_{cross}$ into $E_{within}$. (a) Subgraph with flows on $E_{cross}$ in the first column. (b) Entering flow and outgoing flow are like extra edges linking to source and sink. (c) Purple cut would segment the subgraph into source side and sink side, where extra edges lead to counting two more cuts in the cost. (d) Costs are added to all paths/cuts below the edge linking to the source and are added to all paths/cuts above the edge linking to the sink. (e) The first column of the converted planar DTW graph. (f) The converted GTW subgraph.

in the supplementary. In the following, we will show "Drain" and "Discharge" operations can be implemented in linear time complexity.

### 3.1.2 Linear time implementation of "Drain" component operation

Regarding "Drain" operation, we only need to calculate the min-cut of one active component that links to the sink, then all the excess above the new cut is drained and the part below the new cut is identified as a new active component (Fig.2(b)(c)). As mentioned, the min-cut can be quickly solved by DP on its dual graph. However, with flows on $E_{cross}$, DP cannot be applied directly. The graph is not planar anymore (Fig.3(a)(b)) and its dual graph does not exist. Here we design a linear-time graph conversion strategy as shown in Fig.3 (see algorithms in the supplementary), which can incorporate known flows on $E_{cross}$ into $E_{within}$ and obtain an equivalent planar graph where DP can still be applied. Combining linear conversion strategy with DP, the "Drain" component operation can be implemented in linear time.

**Lemma 2:** The corresponding cut values/path costs before and after graph conversion are the same.

### 3.1.3 Linear time implementation of "Discharge" component operation

To discharge one component, we try to push excess out on each node as much as possible to make sure the component has sent all the excess it can since excess can flow anywhere in the same component.

**Lemma 3:** The amount of maximum possible excess on node $v$ is the min-cut value in residual graph by taking $v$ as sink.

Working on the modified DTW graph, DP can still be applied to calculate such excess by finding the difference between the shortest path below $v$ and the shortest path of the whole graph. The difference can represent the min-cut value mentioned in Lemma 3. To implement linear-complexity component discharge operation, here we use another layer of DP that reuses the stored dynamic matrices that save the shortest distance recursively. The detailed algorithm of "Discharge" is given in the supplementary.

### 3.1.4 A new labeling function to guide excess approaching the sink across components

We design a new labeling function to guide excess approaching $t$ across components from high label to low label. Different from the one in generic push-relabel [9], our labeling function implies the distance from one component to sink rather than from the node. Since the cut blocks the connection of components in the same subgraph, only $E_{cross}$ is counted. Derived from such distance, we define our labeling function $d : V \rightarrow \mathbb{N}$ to be valid if for all residual edges,

$$d(v) \leq d(w) \qquad (v, w) \in E_{within} \tag{2a}$$
$$d(v) \leq d(w) + 1 \quad (v, w) \in E_{cross} \tag{2b}$$

With such definitions, we have,

**Lemma 4:** All nodes in the same component have the same label.

**Lemma 5:** The new labeling function (2) is consistent with generic push-relabel labeling function if treating each component as one unit.

---

**Algorithm 1** ELCO

---

$R_i = G^i$, $d(R_i) = 0$, $cut = \text{Drain}(R_i)$, Split $(R_i, cut)$ for $i = 1, 2, ..., N$      ▷ Initialization
**while** active region exists **do**
    Pick active region with highest label, $R_i$      ▷ Highest-label selection rule
    **if** $d(R_i) = 0$ **then**      ▷ sink component
        $cut = \text{Drain}(R_i)$, Split $(R_i, cut)$      ▷ First type operation and update
    **else**      ▷ No direct edge linking to sink
        **if** $\exists R_j$ that $R_i \rightarrow R_j$ and $d(R_i) = d(R_j) + 1$ **then**
            $cut = \text{Discharge}(R_i)$, Split $(R_i, cut)$      ▷ Second type operation and update
        **else**
            Relabel($R_i$)      ▷ Relabel
            Gap-relabel heuristic      ▷ Relabel may result in label gap
        **end if**
    **end if**
**end while**

---

**Algorithm 2** Relabel($R$), $R \in G^n$

---

$d_{minCross} = \min\{d(R_j)|R \rightarrow R_j, R_j \notin G^n\}$      ▷ Lowest reachable label in neighbor subgraphs
$d_{minWithin} = \min\{d(R_j)|R \rightarrow R_j, R_j \in G^n\}$      ▷ Lowest reachable label in same subgraph
**if** $d_{minCross} < d_{minWithin}$ **then**
    $d(R) = d_{minCross} + 1$
**else**      ▷ Consider the validity
    $d(R) = d_{minWithin}$
    $R = R \bigcup \{R_j | R \rightarrow R_j, d(R) = d(R_j), R \in G^n, R_j \in G^n\}$ into $R$      ▷ Merge
    Relabel($R$)
**end if**

---

Lemma 4 suggests that we can set the same label to one component and all the nodes in it. And lemma 5 implies that ELCO can be seen as an alternative push-relabel approach that uses component as the operation unit, and all the statements in the generic push-relabel approach [9] hold on the component level. Since the validity of the labeling function is maintained in ELCO (see the proof in supplementary), the optimal solution can be achieved in polynomial component operations.

The ELCO is outlined in Algo.1 and Algo.2. Details and an illustrative example can be found in the supplementary. $R$ is the symbol of a component. $R_1 \rightarrow R_2$ if for $v \in R_1$ and $w \in R_2$ that $(v, w)$ has positive residual capacity. Highest-label selection rule is applied, and both global-relabel heuristic and gap-relabel heuristic in [9] are utilized for acceleration.

### 3.2 Bidirectional-pushing strategy

Generic push-relabel methods, and ELCO alike, attempts to push as much as excess to the sink. If part of the excess fails to reach the sink due to the limited edge capacity, they need to be absorbed back by the source, which may take a long time and incur high computational cost. Since this part of excess can neither make an impact on the max-flow nor change the final cut, such computation is redundant. ELCO reduces part of the redundancy within each component through linear component operations, but it cannot solve the problem across components, especially with large $\kappa$ (large capacities of $E_{cross}$). Under large $\kappa$, $E_{within}$ are easier to be saturated and components are more likely to be segmented into a smaller size. Thus, the redundancy is hard to be reduced if we focus on each component alone.

Consider an extreme case in which all nodes are divided into the source side $V_T$ according to the cut. All excess can then reach the sink and all push operations are not redundant. In the other extreme, all nodes belong to the source side $V_S$. Most excess cannot reach the sink but is still pushed, leading to redundant computation. Reversely, rather than pushing excess from the source, a better strategy is pushing deficit from the sink. Deficit means input flow is less than output flow. Pushing deficit is just pushing excess in a graph with source and sink switched. Since all nodes are divided into $V_S$, all operations for pushing deficit in the opposite direction are not redundant. In conclusion, pushing excess in the sink side $V_T$ or pushing deficit in the source side $V_S$ will not result in invalid

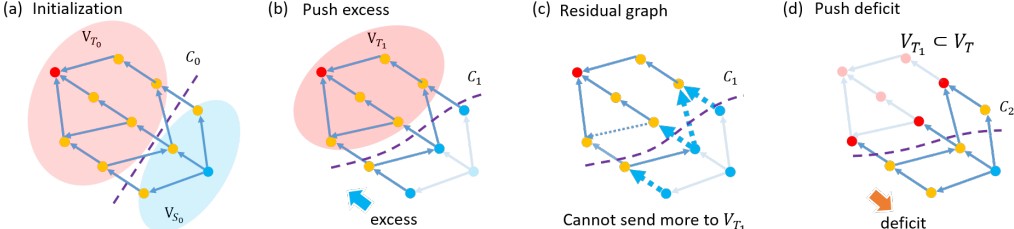

Figure 4: Bidirectional pushing. (a) Initial cut $C_0$ segments graph into $V_{S_0}$ and $V_{T_0}$. (b) Replace nodes in $V_{S_0}$ by source, pushing excess leads to the new cut $C_1$ which segments the new sink side $V_{T_1}$ out. (c) The residual graph of (b), while $V_{S_0}$ cannot push more before replacing nodes. (d) Replace nodes in $V_{T_1}$ by sink, the optimal cut of the whole problem is found by pushing deficit.

computation. If prior knowledge is available to guide us to find an initialization of warping paths, we only need to push excess on the sink side and deficit on the source side. After that, little computation would be wasted. This idea motivates us to design the bidirectional-pushing strategy (Fig.4):

- Initialization: Estimate initial cut $C_0$ for GTW graph. The corresponding source side and sink side are denoted as $V_{S_0}$ and $V_{T_0}$, respectively.
- Push excess: Replace all the nodes in $V_{S_0}$ by source, solve the max-flow problem by pushing excess. The new sink side segmented by the new min-cut $C_1$ is denoted as $V_{T_1}$.
- Push deficit: Replace all the nodes in $V_{T_1}$ by sink, solve the max-flow problem by pushing deficit. The obtained min-cut $C_2$ is the min-cut of the original GTW graph.

The strategy is guaranteed to achieve an optimal solution by the following statements:

**Lemma 6:** Assume the min-cut segments the nodes $V$ into source side $V_S$ and sink side $V_T$, then replacing the nodes in $V_S$ or $V_T$ by source or sink does not impact the min-cut.

**Lemma 7:** Assume $V_T$ represents the nodes of sink side segmented by the real min-cut of original GTW graph, then $V_{T_1} \subseteq V_T$.

**Theorem 1:** The obtained min-cut in bidirectional strategy is the min-cut of the original graph.

In joint alignment problem, since the final cuts tend to be similar because larger $\kappa$ brings larger similarity penalty, a good initialization is easy to be found.

### 3.3 Bidirectional pushing with linear component operations

We propose BILCO as the top-level algorithm integrating ELCO and the bidirectional-pushing strategy, its source code can be found at https://github.com/yu-lab-vt/BILCO. Under small $\kappa$, the component is usually large so that ELCO can work efficiently. While under large $\kappa$, the cuts in different subgraphs are similar and the initial solution can be easily estimated. Therefore, the bidirectional-pushing strategy works well. By setting initialization on each subgraph as the optimal warping path of the averaged subgraph, which is also the solution of joint alignment with infinite $\kappa$, BILCO could take both advantages and has a good performance under any $\kappa$. We use such initialization in our experiments and get good performance. Therefore, we recommend it as the default setting for BILCO, at least for similar applications.

### 3.4 Complexity

Bidirectional-pushing strategy can convert the original problem into two smaller sub-problems, which may help accelerate *any* max-flow methods. For example, in the best case, the initialization is accurate and divides the problem into two half-size sub-problems. Then the speed for one $O(|V|^2\sqrt{|E|})$ method may be accelerated nearly 3 times since both $|V|$ and $|E|$ are halved. Even with a bad initialization, the strategy can only increase the operation number without impacting the complexity bound.

ELCO is somehow like the integration of DTW (with complexity $\Theta(|V|)$) and HIPR (with complexity $O(|V|^2\sqrt{|E|})$ [5]). On one hand, $E_{cross}$ makes ELCO more complex than DTW, thus it has a lower

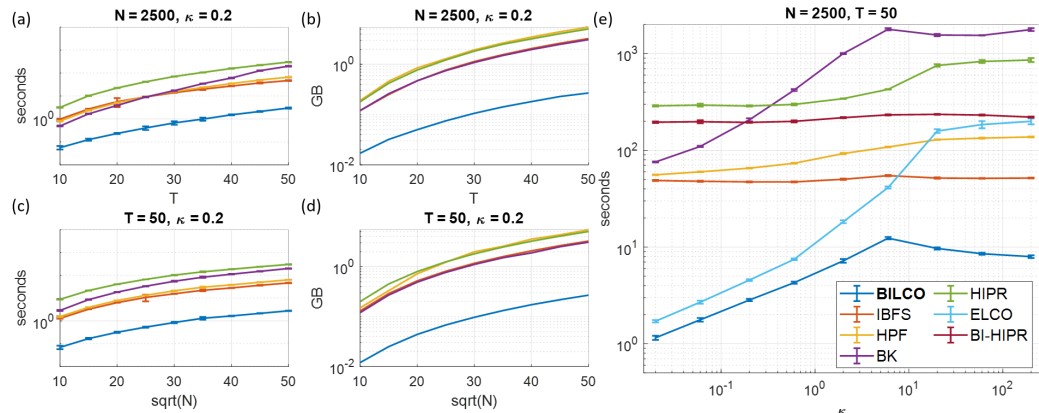

Figure 5: (a)-(d) compare the running time and memory usage under different graph size. (e) compares the running time of max-flow methods under different $\kappa$, including BI-HIPR and ELCO.

bound $\Omega(|V|)$. On the other hand, using linear component operations to push excess, ELCO has fewer operation units and less complexity than HIPR. Even in the worst case where each component contains only one node, ELCO is still equivalent to HIPR with the same upper bound.

Therefore, the worst complexity of BILCO is $O(|V|^2\sqrt{|E|})$.

## 3.5 Memory usage

BILCO can also improve memory efficiency by utilizing the structure of GTW graph. In general max-flow methods, the majority of memory is used to record the relationship between nodes and edges. With a known graph structure, BILCO stores nodes and edges in coordinate order so that the relationship can be derived from their positions. The expense for storing components is insignificant in real applications since the component number is usually neglectable compared with large $|V|$ and $|E|$. Table 1 compares the memory usage of BILCO and peer methods based on their implementation.

Table 1: Memory usage (bytes) among BILCO, IBFS, HPF, BK, and HIPR.

| BILCO | IBFS | HPF | BK | HIPR |
|---|---|---|---|---|
| $8|V| + 4|E|$ | $48|V| + 64|E|$ | $156|V| + 96|E|$ | $48|V| + 64|E|$ | $64|V| + 112|E|$ |

# 4 Experiments

In this section, we compare the efficiency of BILCO with four peer methods IBFS, HPF, BK, and HIPR under synthetic simulation and real applications. All experiments were conducted in MATLAB with Intel(R) Xeon(R) Gold 6140@2.30Hz, 128GB memory, Windows 10 64-bit, and Microsoft VC++ compiler. No GPU is used. All methods are implemented in C/C++ with MATLAB wrapper.

## 4.1 Synthetic data

Following a similar scheme in [22], we simulated an image signal propagation dataset with varying pixels ($N$) and frames ($T$). 4-connected neighbors are used and thus the GTW graph has around $|V| = 2NT^2$ nodes and $|E| = 7NT^2$ edges. A bell-shaped signal propagated from the center of the image to the boundary. Gaussian noise was added so that the signal-to-noise ratio is 10dB, which mimics a real scenario. To make hyperparameter $\kappa$ comparable, we normalized the synthetic data by dividing the standard deviation of the noise. We tested 20 instances for each combination of $N$, $T$, and $\kappa$.

Fig.5 (a-d) compares BILCO and peer methods under different graph sizes, where BILCO is at least 10 times faster than any peer method and costs only $1/10$ memory. The impact of hyperparameter $\kappa$ is demonstrated in Fig.5 (e), where HIPR combined with our bidirectional-pushing strategy (BI-HIPR) and ELCO method are also compared. As shown, the running time of the majority of the methods

Table 2: Efficiency comparison of different methods on real data.

| Problem | | | BILCO | IBFS | HPF | BK | HIPR |
|---|---|---|---|---|---|---|---|
| Name | \|V\| | \|E\| | Time
Memory | Time/(Time of BILCO)
Memory/(Memory of BILCO) | | | |
| Calculate signal propagation in imaging data (first two generated by us, last two from [20] and [21]) | | | | | | | |
| Glia Ca$^{2+}$ | $1.1 \times 10^9$ | $3.8 \times 10^9$ | **586s**
**25.1GB** | Out of
Memory | Out of
Memory | Out of
Memory | Out of
Memory |
| Glia Ca$^{2+}$ * | $7.0 \times 10^7$ | $2.3 \times 10^8$ | **7s**
**1.5GB** | ×50.0
×12.1 | ×109.4
×21.1 | ×348.0
×12.1 | ×385.6
×18.6 |
| 5_rat_astro_ATP | $7.9 \times 10^7$ | $2.7 \times 10^8$ | **32s**
**1.6GB** | ×17.8
×12.8 | ×19.5
×22.4 | ×480.2
×12.8 | ×76.3
×19.8 |
| Exvivo Ca$^{2+}$ | $6.5 \times 10^7$ | $2.2 \times 10^8$ | **26s**
**1.4GB** | ×20.2
×12.1 | ×18.4
×21.1 | ×665.7
×12.1 | ×74.1
×18.6 |
| Extract depth information in binocular stereo (data from [16]) | | | | | | | |
| pendulum1 | $2.9 \times 10^9$ | $7.2 \times 10^9$ | **6065s**
**59.2GB** | Out of
Memory | Out of
Memory | Out of
Memory | Out of
Memory |
| artroom2 | $2.9 \times 10^9$ | $7.2 \times 10^9$ | **7316s**
**58.9GB** | Out of
Memory | Out of
Memory | Out of
Memory | Out of
Memory |
| pendulum1* | $7.7 \times 10^7$ | $1.9 \times 10^8$ | **35s**
**1.9GB** | ×10.0
×7.9 | ×13.6
×16.7 | ×32.5
×7.9 | ×72.6
×14.1 |
| artroom2* | $7.7 \times 10^7$ | $1.9 \times 10^8$ | **33s**
**1.8GB** | ×9.8
×8.3 | ×12.1
×16.7 | ×19.2
×8.3 | ×58.9
×14.8 |
| Signature identification (data from [14]) | | | | | | | |
| Real signatures | $1.7 \times 10^4$
$\sim 6.3 \times 10^6$ | $5.9 \times 10^4$
$\sim 2.2 \times 10^7$ | **204s**
**0.14GB** | ×15.2
×11.4 | ×19.5
×20.7 | ×34.8
×11.4 | ×156.3
×21.4 |
| Forgeries | $1.7 \times 10^4$
$\sim 1.6 \times 10^7$ | $8.8 \times 10^4$
$\sim 5.8 \times 10^7$ | **564s**
**0.53GB** | ×20.9
×8.3 | ×16.6
×14.7 | ×77.9
×8.3 | ×145.4
×14.5 |

grows as $\kappa$ becomes larger, because larger capacities of $E_{cross}$ give more freedom to exchange flow and make the graph more complex. Among all methods, BILCO shows the best efficiency under any $\kappa$. By comparing ELCO with BILCO, or HIPR with BI-HIPR, we find that the bidirectional-pushing strategy speeds up both methods, especially under large $\kappa$. It's because the default initialization in Section 3.3 gets closer to the optimal solution when $\kappa$ gets larger, which effectively removes unnecessary computation. Fig.5(e) also shows that the two strategies, ELCO and bidirectional pushing are complementary and synergistic. When $\kappa$ is small, ELCO dominates the acceleration. When $\kappa$ is large, the bidirectional pushing makes a bigger impact. Interestingly, the two strategies synergistically improve the speed for the large $\kappa$. For example, under the largest $\kappa$, both BI-HIPR and ELCO are nearly 4 times faster than HIPR, suggesting a speedup of 16 folds for independent effects, while BILCO shows around 100-fold speedup compared to HIPR.

## 4.2 Real data

Here we compare our BILCO with four peer methods in three distinct application categories: calculating signal propagation [21], extracting depth information [11], and signature identification [13]. Since all these max-flow methods would give the same results, here we only compare running time and memory usage, as shown in Table. 2, where * in the name represents the spatially downsampled data. BILCO is used as the baseline and we show how many times others cost. The first application is similar to the one in Section 4.1 and we won't go into detail here. In the second application, the depth can be derived from the misalignment of the same row between two images, while the depth distributions in adjacent rows are assumed to be similar. To avoid unnecessary computation while allowing relatively large disparity, we set a window size with $1/5$ sequence length in this experiment. In the third application, 20 persons' signatures [14] are identified where totally 500 real signatures and 500 forgeries are compared with the reference signature. Smoothness is imposed on the feature sequences of the same signature, and in the table we show the summation of running time

and maximum memory usage for two signature categories. The detailed preprocessing steps and experiment settings are given in the supplementary. As shown, BILCO always performs best for all these joint alignment applications. It shows around 10 to 50 times speed improvement and only costs nearly $1/10$ memory compared with the best peer method. Especially for the application of estimating signal propagation, BILCO performs averagely 29 times (range from 18 to 50) faster than other methods. Considering these applications are diverse and the dependency structures are various from 2-connected neighbors for depth, 4-connected neighbors for propagation, to all-connected neighbors for signature, this forcefully demonstrates the superiority and broad applicability of BILCO.

## 5    Conclusion and discussion

Compared with DTW, joint alignment utilizes the dependency in the data and could obtain better performance, however, at the expense of speed and memory. In this paper, we developed BILCO algorithm to minimize such an expense by exploiting the special properties of the problem. It can efficiently solve joint alignment max-flow problems and get exactly the global optimal solution without sacrificing accuracy. Besides theoretical analysis, we demonstrated the efficiency of BILCO through both synthetic experiments and real applications, where BILCO showed around 10 to 50 times speed improvement and only cost averagely $1/10$ memory compared with the best one among peer methods IBFS, HPF, BK, and HIPR. The testing was conducted on a wide range of scenarios, suggesting the broad applicability of BILCO. We expect our work could facilitate greatly the application of joint alignment and help obtain better performance in different fields.

We observed that the bidirectional-pushing strategy can not only be regarded as a general push-relabel acceleration trick but also work synergistically with linear component operations for BILCO. Because of the bidirectional strategy, the graph is separated so that the less unnecessary excess or deficit needs to be pushed, the fewer saturated edges and cuts appear, and then the larger components will be. With larger components, the number of operations can be reduced greatly, which makes BILCO even faster ($100\times$) than the multiplication of BI-HIPR ($4\times$) and ELCO ($4\times$) when compared to HIPR as mentioned in Section 4.1.

Our BILCO algorithm is a specialized approach to a specific max-flow problem. While inventing max-flow algorithms has a long history, all the existing methods fail to solve our problem efficiently, no matter whether they are based on augmenting path [4, 12, 8], localized operations as in push-relabel [9, 5], the combination of the two [17, 7], or integrating graph decomposition [18, 6, 17]. To the best of our knowledge, we are the first of integrating DP and the generic push-relabel method to solve max-flow problems. Although there is a general feeling in the field that initialization does not help much for max-flow algorithms, we showed bidirectional pushing is a good strategy to leverage initialization and it is particularly powerful when coupled with linear component operations. We hope our efforts here will be not only useful for the application of joint alignment but also inspiring for the methodology development of specific network flow problems. We have a general sense that while it is hard to invent better generic algorithms for classical problems, there are a lot of opportunities for more specialized problems and indeed great demands due to the super large scale of recent problems.

## Acknowledgments and Disclosure of Funding

We thank the anonymous reviewers for their helpful suggestions. This work was supported by grants NIH R01MH110504, U19NS123719, and NSF 1750931.

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
