# OpenReview forum: "BILCO: An Efficient Algorithm for Joint Alignment of Time Series"
_NeurIPS.cc/2022/Conference — NeurIPS 2022 Accept_

### Official Review · Reviewer_ycT1 · 2022-07-09

**Rating:** 6
**Confidence:** 4
**Soundness:** 3 good
**Presentation:** 3 good
**Contribution:** 3 good

**Summary:**

This paper investigates the joint alignment of multiple time series. Compared to the existing max-flow method on GTW, the proposed method makes use of two properties of the joint alignment max-flow problem, i.e., joint alignment is a generalization of pairwise alignment, and a coarse approximate solution to joint alignment can be readily estimated, for significantly reducing computational time and space.

**Questions:**

1. Is it reasonable to include the comparison of the alignment performance of the compared methods?
2. Are there any trade-off between the alignment performance and the efficiency of the proposed method?


**Limitations:**

The authors addressed the potential negative societal impact of the work well.

**Strengths And Weaknesses:**

Strengths:
1. The observation of two important properties of the joint alignment max-flow problem.
2. The corresponding designs to the two properties, i.e., Excess pushing with Linear Component Operations (ELCO) that integrates DP and the push-relabel approach, and a bidirectional-pushing strategy to utilize prior knowledge as initialization, are reasonable for improving efficiency.
3. The complexity analysis provides good insights in comparison.
4. The experiments on both synthetic and real datasets indicate the efficiency of the proposed method.

Weakness:
1. The experiments focus on comparing time and space efficiency of different methods. Alignment accuracy could be compared to indicate whether there is a trade-off between effectiveness and efficiency of the proposed method.

---

> ### Author Response · Authors · 2022-08-01
> **Rebuttal to Reviewer ycT1**
>
> Thank you for your valuable feedback. The major concern was **whether the high efficiency of our method is at the expense of alignment accuracy.** We would like to take this opportunity to relieve your concern.
>
> First, **the compared peer methods and our proposed method BILCO can achieve the same exact result for a given joint alignment problem.** They are all max-flow algorithms and can achieve the same global optimal solution, thus there is no need to compare the alignment performance between those methods. Sorry for the lack of clarity, we will stress this point in our paper.
>
> Second, **there is no trade-off between the accuracy and efficiency of the proposed method.** The improvement of time and space efficiency is from utilizing the specific characteristics of joint alignment max-flow problem and avoiding redundant operations, rather than getting an approximate solution at the expense of alignment performance.
>
> Besides, here we may need to clarify why we compared the efficiency in Fig. 5 with respect to hyperparameter $\kappa$. $\kappa$, as mentioned in equation (1) and experiments, determines the assumed amount of dependency in the structural data and different $\kappa$ may lead to different alignment accuracies in different applications. It may also impact the characteristics of the joint alignment max-flow problem and thus influence the efficiency of our proposed method (slightly). However, for the same given problem, that is, with the same hyperparameter $\kappa$, all max-flow methods compared in the paper will lead to the same exact result.

---

### Official Review · Reviewer_JqmR · 2022-07-11

**Rating:** 7
**Confidence:** 1
**Soundness:** 3 good
**Presentation:** 3 good
**Contribution:** 3 good

**Summary:**

This paper proposes an algorithm BILCO for solving graphical time warping, an alignment method for multiple time series data. The authors realized a fast and memory-saving algorithm by focusing on the special characteristics of DTW graphs. Experimental results, including results for real data, show that their two proposed methods, ELCO and bidirectional pushing, dramatically reduce computation and memory consumption.

**Questions:**

- Are there any other applications of the proposed method other than graphical time warping?

typo
- l.170: source side $V_T$

**Limitations:**

This paper proposes an algorithm for solving time series data alignment, which does not involve negative social impact.

**Strengths And Weaknesses:**

- The authors propose an algorithm for joint alignment that is much faster (10x~) and memory-saving than other methods.
- Experimentally, the proposed method is applied to three real datasets that require multiple alignment, and shows impressive speedup and memory-saving performance.


- The proposed method is not for general multiple time series alignment problem, but is an algorithm for solving a narrower problem, graphical time warping, in which warping paths in given neighbor relationships should be close, as written in Eq.(1). The application of the proposed method to general multiple alignment problem and other problems is not discussed.

---

> ### Author Response · Authors · 2022-08-01
> **Rebuttal to Reviewer JqmR**
>
> Thank you for your valuable feedback. The major concern was **whether our proposed method can only be applied to graphical time warping (GTW) and thus might be limited in the application scope.** We would like to relieve your concern from three aspects.
>
> First, **our proposed method was indeed designed to solve GTW problems, but GTW itself is flexible, broadly applicable, and can be used as a building block in solving many problems while utilizing structural information.** Its assumed neighbor structure and similarity strength can be application-specific or user-designed, including the special case without any neighbor relationship at all. With the flexible capability of utilizing the structural information, GTW can be integrated into more complex pipelines and applications than the original formulation may suggest. We will also stress this point in our paper.
>
> Second, **there is already some work that extended GTW to more general multiple alignment problems.** A typical application, called neighbor-wise compound-specific graphical time warping (ncGTW) [R1], can achieve better performance for aligning liquid chromatograph-mass spectrometry (LC-MS) profiles **without predefined reference.** Its formulation shares similar characteristics with GTW and our method should also be applicable.
>
> Thirdly, **the idea of our proposed method could be used to accelerate other max-flow techniques.** Although BILCO focuses on the GTW problem, the idea of using component push-relabel operation can be applied to other problems with structural subgraphs. And the bidirectional-pushing strategy can help other push-relabel-based methods with given initialization, which is already shown in Fig. 5(e) (BI-HIPR, combining the strategy with HIPR method).
>
> [R1] Wu, Chiung-Ting, et al. "Targeted realignment of LC-MS profiles by neighbor-wise compound-specific graphical time warping with misalignment detection." Bioinformatics 36.9 (2020): 2862-2871.

---

### Official Review · Reviewer_dysU · 2022-07-11

**Rating:** 6
**Confidence:** 4
**Soundness:** 3 good
**Presentation:** 3 good
**Contribution:** 3 good

**Summary:**

This paper improves the computation speed and memory cost of GTW [Wang, et al, 2016] method, for the multiple time-series joint alignment problem. Two major enhancements are introduced. The first is excess pushing with linear component operations, which treats components (the number is much less than nodes) as the basic unit. The second is a bi-directional pushing (push excess and push deficit), where pairwise DTW initialization is used.
Comprehensive experiments on both synthetic and real world datasets are conducted to evaluate the speed and memory improvement of the proposed method.


**Questions:**

See above and I am willing to hear more opinions.


**Strengths And Weaknesses:**

Strengths:
1. The joint alignment problem is of great importance and interest to the community. The GTW method converts the alignment problem into a max-flow problem that sheds the light on solving it optimally via techniques in graph theory. This paper further pushes this direction to speed up the process and reduce the memory cost significantly. This makes the GTW approach feasible in real world applications.
2. The paper has illustrative figures to demonstrate the approach.
3. Time and space complexity analysis is provided.
4. Several max-flow algorithms are included in the experiments part. The proposed method performs consistently much better than those baselines.

Weakness:
1. The paper lacks some introduction / background on graph techniques, making it slightly difficult to follow especially when defining ‘drain’, ‘discharge’ and ‘component’. With an illustrative example that demonstrates how these operations are done, would significantly improve the readability of the paper.
2. The most critical problem lies in the definition of GTW. In GTW, the joint alignment problem is converted to a slightly different problem: constraint the similarity of pairwise warping functions / paths. The warping function could vary a lot, but the effect could be very similar. Let me use edit distance to illustrate (slightly different than DTW, but shares similar DP spirits, so I use this as an easy example). Given a reference string, there is one string that needs prepending a letter to match the reference (this is one warping). There is another string that needs appending a letter to match the reference (another warping). These two warping / editing functions are very different, but their integral is the same (both accumulate to only one insertion edit), so their effects are also similar as well. Therefore, some external metric, like the integral of the warping function, their difference should be considered as the penalty, instead of the function itself. GTW makes this definition for ease of computation via max-flow algorithms, but lacks the practical meaning as it is often hard to interpret the warping functions’ difference, and hard to apply to practical tasks like finding the centroids of time series. I think this formulation reduces the interests of GTW based approaches to the community.

---

> ### Author Response · Authors · 2022-08-01
> **Rebuttal to Reviewer dysU**
>
> Thank you for your valuable and insightful feedback. The major concerns were: **1) the lack of introduction to graph techniques and an illustrative example makes our graph operations hard to understand and 2) the penalty for warping function dissimilarity in graphical time warping (GTW) lacks the practical meaning and may weaken the interests of our method.** We would like to take this opportunity to relieve these concerns.
>
> First, **we will provide a more detailed introduction to the relevant graph techniques and add an illustrative example to demonstrate our graph operations.** Your suggestion of an illustrative example is really appreciated. We will incorporate it into our revision. Moreover, we will also provide a vivid animation to demonstrate our approach as supplementary material.
>
> Second, **joint alignment formulations with different external metrics have different focuses and advantages in different fields. In many applications, the warping functions have practical meanings and the penalty for warping function dissimilarity is meaningful.** For example, in our first example application (Fig. 1(e)), calculating signal propagation, the warping function represents the delay of the real signal at each time point relative to the reference time series. Thus, if two signals propagate similarly, the two corresponding warping functions should be close, while the same integral of the warping function could not assure a similar propagation. Or in our second example application (Fig. 1(f)), extracting depth in binocular stereo, the warping function represents the disparity between two views and can be used to derive the depth. It is natural to assume the depths of neighbor locations to be similar. In those applications, the structural information can be utilized to improve alignment performance by adding constraints on warping function dissimilarity. We will stress this point in our paper. Besides, we agree that in many applications the joint alignment should add constraints on some other external metrics (e.g., the integral of warping function, as you suggest). Thank you for pointing it out. We will consider other external metric-based similarities and extend our method in future work.

---

> > ### Comment · Reviewer_dysU · 2022-08-10
> > **Thanks for response**
> >
> > Thanks for the response and hope to see the illustrative example in the final version.

---

### Meta-Review · Area_Chair_wtxs · 2022-08-23

**Recommendation:** Accept
**Confidence:** Certain

**Metareview:**

In this paper, the authors propose an algorithm BILCO for solving graphical time warping, an alignment method for multiple time series data. Overall, the proposed approach is interesting, and all reviewers are positive. Thus, I also vote for acceptance.

**Award:**

No

---

### Decision · Program_Chairs · 2022-09-14

Accept